# Novel Green Fluorescent Polyamines to Analyze ATP13A2 and ATP13A3 Activity in the Mammalian Polyamine Transport System

**DOI:** 10.3390/biom13020337

**Published:** 2023-02-09

**Authors:** Marine Houdou, Nathalie Jacobs, Jonathan Coene, Mujahid Azfar, Roeland Vanhoutte, Chris Van den Haute, Jan Eggermont, Veronique Daniëls, Steven H. L. Verhelst, Peter Vangheluwe

**Affiliations:** 1Laboratory of Cellular Transport Systems, Department of Cellular and Molecular Medicine, KU Leuven, B-3000 Leuven, Belgium; 2Aligning Science Across Parkinson’s (ASAP) Collaborative Research Network, Chevy Chase, MD 20815, USA; 3Laboratory of Chemical Biology, Department of Cellular and Molecular Medicine, KU Leuven, B-3000 Leuven, Belgium; 4Leuven Viral Vector Core, KU Leuven, B-3000 Leuven, Belgium; 5Research Group for Neurobiology and Gene Therapy, Department of Neurosciences, KU Leuven, B-3000 Leuven, Belgium; 6Leibniz Institut für Analytische Wissenschaften ISAS, e.V., 44227 Dortmund, Germany

**Keywords:** fluorescently labeled polyamines, mammalian polyamine transport systems, P_5B_-type ATPases, radiolabeled polyamines

## Abstract

Cells acquire polyamines putrescine (PUT), spermidine (SPD) and spermine (SPM) via the complementary actions of polyamine uptake and synthesis pathways. The endosomal P_5B_-type ATPases ATP13A2 and ATP13A3 emerge as major determinants of mammalian polyamine uptake. Our biochemical evidence shows that fluorescently labeled polyamines are genuine substrates of ATP13A2. They can be used to measure polyamine uptake in ATP13A2- and ATP13A3-dependent cell models resembling radiolabeled polyamine uptake. We further report that ATP13A3 enables faster and stronger cellular polyamine uptake than does ATP13A2. We also compared the uptake of new green fluorescent PUT, SPD and SPM analogs using different coupling strategies (amide, triazole or isothiocyanate) and fluorophores (symmetrical BODIPY, BODIPY-FL and FITC). ATP13A2 promotes the uptake of various SPD and SPM analogs, whereas ATP13A3 mainly stimulates the uptake of PUT and SPD conjugates. However, the polyamine linker and coupling position on the fluorophore impacts the transport capacity, whereas replacing the fluorophore affects polyamine selectivity. The highest uptake in ATP13A2 or ATP13A3 cells is observed with BODIPY-FL-amide conjugated to SPD, whereas BODIPY-PUT analogs are specifically taken up via ATP13A3. We found that P_5B_-type ATPase isoforms transport fluorescently labeled polyamine analogs with a distinct structure–activity relationship (SAR), suggesting that isoform-specific polyamine probes can be designed.

## 1. Introduction

Polyamines such as putrescine (PUT), spermidine (SPD) and spermine (SPM) are ubiquitous and physiologically important organic polycations found in every living cell. Polyamines are implicated in a broad range of cellular processes, ranging from cell proliferation to signaling, but their levels decline with aging [1,2]. Conversely, polyamine supplementation increases the lifespan of model organisms such as mice and fruit flies [3,4]. At the molecular level, polyamine biosynthesis and catabolism pathways are well understood, but a clear knowledge gap remains regarding the molecular characteristics of the mammalian polyamine transport system (mPTS). Three main mechanisms have been proposed to be involved in cellular polyamine uptake [5]: (i) a direct transport at the plasma membrane, (ii) a glypican-mediated uptake via endocytosis and (iii) a caveolin-1-mediated endocytosis [5,6,7,8,9,10]. Recently, we identified two key players of the mPTS that belong to the P_5B_-type ATPases (ATP13A2 and ATP13A3) [11,12,13].

In humans, ATP13A2 and ATP13A3 are ubiquitously expressed with highest expression in the brain and the liver, respectively ([14] and https://www.proteinatlas.org accessed on 6 February 2023). At the subcellular level, both proteins are localized in the endosomal pathway with a preferred late endo-/lysosomal distribution for ATP13A2 [12,13,14] and early/recycling endosomes for ATP13A3 [12,13,15]. We provided biochemical evidence that human ATP13A2 (hATP13A2) transports polyamines from late endo/lysosomes to the cytosol with a high affinity for SPM and SPD [13], which was confirmed at the structural level [16,17,18,19,20]. ATP13A2, and also the closely related ATP13A3, fulfill their polyamine transport function downstream of polyamine internalization via endocytosis [11,13]. Both transporters are implicated in disease, since genetic mutations in *ATP13A2* have been linked to neurodegeneration, whereas *ATP13A3* is genetically implicated in pulmonary arterial hypertension and may play a role in cancer (see [14] and references therein). Studying the transporters of the mPTS in cells typically relies on the use of radiolabeled (^3^H and/or ^14^C) or fluorescently labeled polyamines. We recently reported fluorescent polyamine conjugates incorporating a symmetrical BODIPY fluorophore (referred to as ‘BODIPY’ throughout the manuscript). These have proven invaluable to characterize the P_5B_-type ATPases [11,21]. Indeed, ATP13A2 contributes to the cellular uptake of BODIPY-SPM and BODIPY-SPD in human neuroblastoma (SH-SY5Y) cells [13] corresponding to the substrate specificity of purified ATP13A2 for SPM and SPD [13]. We further demonstrated that ATP13A3 is responsible for the impaired BODIPY-PUT uptake observed in the CHO-MG cell line with a deficient mPTS [11,22].

Despite their proven value, it remains unclear how the transport of these polyamine conjugates compares with unlabeled polyamines and whether the fluorophore or its linkage to the polyamine may influence the recognition and transport by the P_5B_-type ATPases. Here, we report our recent work aimed at addressing these questions. Specifically, we compared uptake of BODIPY- and ^14^C-labeled polyamines in two cell models relying either on ATP13A2 or ATP13A3 activity. Additionally, with a series of new green fluorescent analogs of PUT, SPD and SPM we tested the impact of different coupling strategies and fluorescent headgroups on transport capacities. Interestingly, we found that a broad range of fluorescent polyamine probes are taken up in cells via the catalytic activity of ATP13A2 and/or ATP13A3, although probe- and isoform-specific effects were observed. Overall, this work provides novel fluorescent polyamine conjugates with improved properties for the study of P_5B_-type ATPases in disease models, and opens the way for the design of P_5B_ isoform-specific polyamine probes.

## 2. Materials and Methods

### 2.1. Preparation of Compounds

All fluorescent polyamine probes were prepared in a final stock concentration of 5 mM in 0.1 M γ-(N-Morpholino) propanesulfonic acid (MOPS; PanReac AppliChem, Chicago, IL, USA, A1076,1000), brought to pH 7.0 with KOH (Honeywell, Mecklenburg County, NC, USA, 319376 Fluka) and stored at −20 °C. Unclicked SPM-N_3_-4·HCl (azide SPM) was dissolved in MOPS-KOH to reach 20 mM, unclicked BODIPY-alkyne was dissolved in dimethyl sulfoxide (DMSO; Sigma, St. Louis, MI, USA, D5879) to a final concentration of 5 mM and they were both stored at −20 °C. Unlabeled polyamines, putrescine dihydrochloride (PUT; Sigma: P7505), spermidine (SPD; Sigma: S2626) and spermine (SPM; Sigma: 85590) were dissolved in MOPS-KOH (pH 7.0) at a final concentration of 500 mM (putrescine and spermidine) or 200 mM (spermine) and stored at −80 °C.

For the biochemical ATP/NADH-enzyme coupled ATPase assay, the following stock solutions were prepared: 2.5 M potassium chloride (KCl; Sigma: P5405), 1 M magnesium chloride solution (MgCl_2_; Gibco (Waltham, MA, USA): M1028), 15.625 U/µL lactate dehydrogenase (Sigma: L2500) and 4 U/µL pyruvate kinase (Sigma: P1506). They were stored at 4 °C. Additionally, 100 mM β-nicotinamide adenine dinucleotide reduced disodium salt hydrate (NADH; Sigma: 43420) in MOPS-KOH (pH 7.0), 50 mM phospho(enol)pyruvic acid tri(cyclohexylammonium) salt (PEP; Sigma: P7252), 500 mM dithiothreitol (DTT; PanReac AppliChem: A2948.0025) and 63 mM adenosine 5′-triphosphate disodium salt trihydrate brought to pH 7.0 (Roche (Basel, Switzerland): ATPD-RO) were stored at −20 °C.

For the preparation of HMEC-1 cell culture medium, a 100 µg/mL stock of human epidermal growth factor (hEGF; Sigma: E9644) was prepared in 0.22 µm filter-sterilized 10 mM acetic acid (Chem Lab (Nairobi, Kenya): CL00.0119.1000), supplemented with 1% albumin fraction V (Carl Roth (Karlsruhe, Germany): 8076.4), in Milli-Q, and was stored at −80 °C in aliquots. Additionally, hydrocortisone (Sigma: H0888) was dissolved in absolute ethanol (VWR Chemicals (Radnor, AR, USA): 20821.296), diluted 1:25 in MCDB131 medium to a final concentration of 100 µg/mL and stored in aliquots at −20 °C.

The endocytosis inhibitors dynasore (Sigma: D7693), genistein (Abcam (Cambridge, UK): ab120112), and pitstop2 (Sigma: SML1169) were dissolved in DMSO to a final stock concentration of 50 mM, 25 mM and 25 mM, respectively, and stored at −20 °C.

### 2.2. Lentiviral Transduction and Cell Culture

SH-SY5Y human immortalized neuroblastoma cells (ATCC: CRL-2266™, Lot Number 62431864; RRID: CVCL_0019) were transduced with lentiviral vectors to obtain stable overexpression of human ATP13A2 (isoform 2, wild-type (ID: NP_001135445) or catalytically dead mutant, D508N) [13]. Lentiviral vectors were produced by the Leuven Viral Vector Core using pLenti HsATP13A2 WT (Addgene plasmid #171485; http://n2t.net/addgene:171485 (accessed on 6 February 2023); RRID: Addgene_171485) as described in dx.doi.org/10.17504/protocols.io.bw57pg9n. SH-SY5Y cells were cultured in DMEM high glucose culture medium (Gibco: 41965), supplemented with 1% MEM Non-Essential Amino Acid Solution (Merck (Rahway, NJ, USA): M7145), 1 mM sodium pyruvate (Gibco: 11360070), 1% Penicillin-Streptomycin (Sigma: P4458) and 15% heat-inactivated Fetal Bovine Serum Standard, South America origin, (FBS; PAN BioTech (Aidenbach, Germany): P30-3306) at 37 °C with 5% CO_2_. After lentiviral transduction, cells were selected with 2 µg/mL puromycin (Invivogen (San Diego, CA, USA): ant-pr-1).

HMEC-1 human immortalized microvascular endothelial cells (ATCC: CRL-3243™, Lot Number: 70022309; RRID: CVCL_0307) were transduced with lentiviral vectors to obtain stable overexpression of human ATP13A3 (wild type or catalytically dead mutant, D498N). Lentiviral vectors were produced by the Leuven Viral Vector Core (Addgene IDs: ATP13A3 WT, 195849; ATP13A3 D498N, 195850; RRID: Addgene_82067). HMEC-1 cells were cultured in 0.2% gelatin (Sigma, G1393) coated 75 cm^2^ flasks and in MCDB131 culture medium without L-glutamine (Gibco: 10372-019), supplemented with 10 ng mL-1 hEGF, 1 µg mL-1 Hydrocortisone, 10 mM GlutaMAX supplement (Gibco: 35050061), 1% Penicillin-Streptomycin and 10% heat-inactivated FBS Standard, South America origin, at 37 °C with 5% CO_2_. After lentiviral transduction, cells were selected with 1 µg/mL puromycin. Note that we used heat-inactivated FBS to deplete the polyamine oxidase activity, which may affect the fluorescent polyamine probes. All cell lines were routinely tested for mycoplasma contamination using the PlasmoTest Mycoplasma detection kit (Invivogen: rep-pt1).

### 2.3. Western Blotting

Cells were detached either by scraping them in Dulbecco’s phosphate-buffered saline modified without calcium chloride and magnesium chloride (DPBS; Gibco: D8537) (SH-SY5Y) or using 0.25% Trypsin-EDTA (Gibco: 25200056) (HMEC-1), for which the enzymatic reaction was stopped by the addition of culture medium. Cell suspensions were centrifuged at 4 °C for 5 min at 450 g (SH-SY5Y) and 2500 rpm (HMEC-1). Cell pellets were washed twice with DPBS and centrifuged again before being lysed in RIPA buffer (RIPA lysis and extraction buffer (Invitrogen (Waltham, MA, USA): 89900) supplemented with protease cocktail inhibitors (SIGMAFAST Protease Inhibitor Cocktail Tablets, EDTA-Free (Sigma: S8830)). Lysis was performed on ice for 30 min before a further centrifugation at 4 °C for 30 min at 20,000 g. Supernatants were kept and protein concentration was estimated using the micro-BCA Protein Assay Kit (Pierce BCA Protein Assay Kit (Thermo Scientific (Waltham, MA, USA): 23225)). A total of 20 µg of protein was mixed with NuPAGE LDS sample buffer (Invitrogen: NP0007) with 5% β-mercaptoethanol. Samples were not boiled, separated on 4–12% Bis-Tris gels (Invitrogen: NP0321BOX) and transferred onto PVDF membranes (Thermo Scientific: 88518) using a liquid transfer (1h15, 100V, 4 °C). Membranes were then blocked in blocking buffer (5% milk powder in 1X TBS and 0.1% Tween20) for 1 h at room temperature, then incubated overnight at 4 °C with the primary antibodies 1% bovine serum albumin in 1X TBS-Tween20 (TBS-T) buffer and washed three times for 5 min in TBS-T. Membranes were then incubated with peroxidase-conjugated secondary goat anti-rabbit or goat anti-mouse antibodies in 1% milk powder in 1X TBS-T buffer for 1 h at room temperature and later washed five times for 5 min in TBS-T. Signal was detected with chemiluminescence reagent (SuperSignal West Pico PLUS chemiluminescent Substrate, Thermo Scientific: 34095) using a Biorad camera (Vilber Lourmat, Collégien, France) and its software (ImageLab). This specific protocol for ATP13A2 and ATP13A3 Western blotting is also described in protocols.io (dx.doi.org/10.17504/protocols.io.81wgbyzqovpk/v1).

Mouse monoclonal anti-GAPDH (Sigma-Aldrich Cat# G8795, lot #067M4785V, RRID:AB_1078991, dilution 1:5000) and rabbit anti-ATP13A2 antibodies (Sigma-Aldrich Cat# A3361, lot #0000102992, RRID:AB_10597403, dilution 1:1000) were purchased from Sigma. Goat anti-rabbit IgG (H+L) secondary antibody HRP conjugated (Thermo Fisher Scientific Cat# 31460, RRID:AB_228341, dilution 1:2000) and goat anti-mouse IgG (H+L) secondary antibody HRP conjugated (Thermo Fisher Scientific Cat# 31430, RRID:AB_228307, dilution 1:2000) were from Thermo Fisher Scientific, and rabbit anti-ATP13A3 antibody (Atlas Antibodies Cat# HPA029471, lot # 000035781, RRID:AB_10600784, dilution 1:2000) was from Atlas Antibodies.

### 2.4. ATP/NADH-Enzyme Coupled ATPase Assay

Purified human ATP13A2 protein was obtained as previously described in [13]. Protein concentration was determined using a Pierce 660 nm Protein Assay (Thermo Scientific: 22660), and the quality of the purification was assessed via SDS-PAGE followed by InstantBlue Coomassie protein stain (abcam: ab119211) staining and immunoblotting, as described in [11]. To measure ATPase activation of purified ATP13A2, serial dilutions of the unlabeled, unclicked azide spermine, BODIPY-SPM, and unclicked alkyne BODIPY were prepared ranging from 0.01 µM to 10 mM, in a final volume of 25 µL per well in a 384-well clear polystyrene microplate. Then, 40 µL of the reagent mix containing 50 mM MOPS-KOH (pH 7.0), 100 mM KCl, 30 mM MgCl_2_, 0.6 mM NADH, 1.667 mM PEP, 2.4 U µL-1 pyruvate kinase, 2.4 U µL-1 lactate dehydrogenase and 2 mM DTT, in the presence or absence of 1.25 µg purified ATP13A2 protein, was added per well. Next, 5 mM ATP (pH 7.0) was added in each well to start the biochemical reaction. The plate was shaken for 30 s prior to acquisition. Absorbance at 340 nm was measured at RT every 30 s for 1 h with a SpectraMax Plus 384 microplate reader (Molecular Devices). Data analysis was conducted using SoftMax Pro 7 (http://www.moleculardevices.com/systems/microplate-readers/softmax-pro-data-acquisition-and-analysis-software (accessed on 6 February 2023); RRID:SCR_014240) and GraphPad Prism 9.3.1 software (http://www.graphpad.com/ (accessed on 6 February 2023); RRID:SCR_002798). This protocol is further described in protocols.io following dx.doi.org/10.17504/protocols.io.6qpvr4do2gmk/v1.

### 2.5. Chemical Synthesis of Fluorescently Labeled Polyamines

The detailed synthesis of each compound is further described in Appendix A.

### 2.6. Acquisition of Absorption and Emission Spectra

Absorbance spectra were determined using the Cary 60 UV-Vis spectrophotometer with 1 nm spectral resolution. Attenuators were used to remove the effect of background and noise. Emission spectra were measured using an Edinburgh Instruments FLS 980 spectrometer at an excitation wavelength of 465 nm and emission data were collected at 1 nm intervals. For both emission and absorbance measurements, probe samples were dissolved in deionized water to keep the optical density below 0.2. After this, the samples were added to a quartz cuvette with 10 mm pathlength before being sealed and measured.

### 2.7. Cellular Polyamine Uptake and Endocytosis Assay

Cells were seeded in 12-well plates to reach 70% confluency the day of the experiment (approximately 2.0 × 10^5^ and 1.5 × 10^5^ cells per well, for SH-SY5Y and HMEC-1, respectively). To assess the polyamine uptake capacity, cells were incubated with 0.1 µM to 100 µM of fluorescently labeled polyamines in their respective medium for 0 to 16 h at 37 °C and 5% CO_2_. To assess endocytosis rate, cells were treated with a cocktail of endocytosis inhibitors containing 100 µM dynasore, 50 µM genistein and 50 µM pitstop2; dissolved in their respective culturing medium without supplementation of FBS for 30 min at 37 °C and 5% CO_2_ and kept at 4 °C for 15 min. Then, cells were co-incubated with 50 µg/mL Alexa647-Transferrin (Invitrogen: T23366) for 20 min at 4 °C, and further incubated for 20 min at 37 °C and 5% CO_2_. After either treatment, cells were washed with DPBS or Versene Solution (Gibco: 15040), detached with either TryplE or 0.25% Trypsin-EDTA, centrifuged and resuspended in DPBS containing 1% Albumin Fraction V. Cell suspensions were filtered through a nylon filter to avoid clumps and kept on ice before acquisition using a BD FACS Canto II AIG or HTS Flow Cytometer (BD Biosciences). A 488 nm 20 mW Solid State Blue (Coherent) laser with 530/30 BP detector and a 633 nm 17 mW HeNe Red (JDS Uniphase) laser with 660/20 detector were used to record the mean fluorescent intensities (MFI) of 10,000 events per sample. These protocols are described in protocols.io following dx.doi.org/10.17504/protocols.io.n92ldp8qxl5b/v1 and dx.doi.org/10.17504/protocols.io.8epv5jjedl1b/v1, respectively. Data analysis was conducted using Flowing Software 2.5.1 (https://en.freedownloadmanager.org/Windows-PC/Flowing-Software-FREE.html (accessed on 6 February 2023); RRID: SCR_015781) and GraphPad Prism 9.3.1.

### 2.8. Radiolabeled Polyamine Uptake Assay

Cells were seeded in 12-well plates to reach 70% confluency the day of the experiment. Cells were incubated with 0.5 to 5 µM of [^14^C]-radiolabeled polyamines ([^14^C]-PUT: ARC 0245-50 µCi; [^14^C]-SPD: ARC 3138-50 µCi; and [^14^C]-SPM: ARC 3139-50 µCi) in culture medium for 30 min at 37 °C. Afterwards, cells were washed with cold DPBS, supplemented with 50 µM of the respective unlabeled polyamine. After two washing steps with cold DPBS, the cells were lysed using 0.1% SDS (Sigma, 71725) in DPBS. After 10 min, the cell lysates were scraped off the wells and collected into scintillation vials containing 7 mL EcoLite Liquid Scintillation Cocktail (MP Biomedicals: 01882475-CF). [^14^C] radioactivity in counts-per-minute (CPM) was measured with liquid scintillation counting (TRI-CARB 4910TR V Liquid Scintillation Counter, PerkinElmer). This protocol is further described in protocols.io following dx.doi.org/10.17504/protocols.io.yxmvm2x85g3p/v1.

### 2.9. Statistics and Data Analysis

Experiments on the different cell models were executed by different researchers, which provided consistent results that independently confirmed the major conclusions. Data are expressed as the mean ± SD, or with individual data points (replicates of multiple independent experiments). GraphPad Prism 9.3.1 software was used to plot all graphs and to perform all of the required statistical assessments. Statistical tests for each graph are described in the legend together with the number of independent biological experiments. For the quantification of immunoblots, ImageJ (https://imagej.net/ (accessed on 6 February 2023); RRID: SCR_003070) was used.

## 3. Results

### 3.1. BODIPY-Conjugated SPM Is a Genuine Transport Substrate of ATP13A2

hATP13A2 exhibits the highest affinity for SPM [13], which binds to a narrow channel-like substrate binding site at the luminal side of the protein [16,17,18,19,20]. Whether the polyamine pocket may also accommodate the bulkier BODIPY-SPM for subsequent transport remains unclear. Since SPM transport by ATP13A2 is coupled to the hydrolysis of ATP, we examined whether BODIPY-SPM also stimulates ATP turnover, and hence represents a genuine substrate. To this end, we purified human WT ATP13A2 (Figure 1A,B) and performed an ATP/NADH enzyme-coupled ATPase assay with SPM or BODIPY-SPM (Figure 1C). ATP13A2′s ATPase activity is stimulated with either SPM, BODIPY-SPM or clickable azide SPM (an intermediate in the coupling strategy), but not with clickable BODIPY-alkyne (Figure 1C). Hence, ATP13A2 recognizes the BODIPY-labeled or clickable SPM as genuine substrates. Surprisingly, ATP13A2 displays a higher apparent affinity, and slightly lower maximal velocity (V_max_), for BODIPY-SPM (K_m_ 0.22 µM) rather than SPM (K_m_ 79 µM).

We previously studied ATP13A2-dependent BODIPY-SPM uptake in human neuroblastoma SH-SY5Y cells, a frequently used cell line in Parkinson’s disease research [13,23]. Here, we characterized the time- and dose-dependency of BODIPY-SPM uptake in SH-SY5Y cells overexpressing ATP13A2 wild type (A2 WT-OE) or a transport-dead mutant (D508N-OE). Both cell models exhibit comparable ATP13A2 protein expression levels (Appendix A) and, interestingly, a significantly higher endocytosis rate in A2 D508N-OE cells compared with A2 WT-OE and non-transduced (NTS) cells (Appendix A), suggesting that the difference in BODIPY-SPM uptake between A2 WT-OE and A2 D508N-OE cells can be attributed to ATP13A2 transport activity. The time-dependency was evaluated by incubating the cells with 5 µM BODIPY-SPM (a concentration that was used before; [13]) up to 16 h (Appendix A). The maximal fold difference in BODIPY-SPM uptake between A2 WT-OE and A2 D508N-OE cells was reached at 4 h (1.6-fold) (Appendix A), whereas maximal cellular uptake capacities were still not achieved at 16 h (Appendix A). We selected a 2 h incubation time, which falls within the linear phase of the BODIPY-SPM uptake (Figure 2A), to examine the dose-dependency of BODIPY-SPM uptake in ATP13A2 cell models (Figure 2B and Appendix A). We observed a five-fold higher BODIPY-SPM uptake in A2 WT-OE versus A2 D508N-OE cells at the lowest BODIPY-SPM concentration (0.1 µM, Appendix A), but the fold difference decreased at higher concentrations. Nevertheless, a significant and reproducible 1.6-fold higher uptake was observed at 5 µM BODIPY-SPM (Figure 2C), which was used in previous studies [13]. Finally, we compared BODIPY-SPM with radiolabeled spermine (^14^C-SPM) uptake in A2 WT-OE cells to assess the impact of the fluorescent tag on cellular uptake capacities (Figure 2D). After incubating A2 WT-OE and D508N-OE cells for 30 min with either 5 µM of BODIPY-SPM or ^14^C-SPM, we observed a significant 1.33-fold higher uptake for ^14^C-SPM (Figure 2D). The effect of the fluorescent tag on SPM uptake reflects the higher apparent affinity, but lower V_max_ of ATP13A2 for BODIPY-labeled versus native spermine (Figure 1C). Overall, our data demonstrate that the BODIPY-SPM probe is a valuable tool to evaluate the ATP13A2 transport activity in cells.

### 3.2. Evaluation of BODIPY-PUT and BODIPY-SPD Uptake in a New Human ATP13A3 Cell Model

Besides ATP13A2, we previously reported that ATP13A3 also modulates cellular BODIPY-polyamine uptake in CHO cells, with the strongest impact on BODIPY-PUT uptake [11]. Here, we decided to evaluate the uptake of the BODIPY-PUT and BODIPY-SPD probes in the context of human ATP13A3, and turned to immortalized human dermal microvascular endothelial cells (HMEC-1). HMEC-1 cells are frequently used in the context of pulmonary arterial hypertension, a disease that is associated with ATP13A3 mutations [24,25,26,27], and do express ATP13A3. We generated HMEC-1 cells with stable lentiviral mediated over-expression of ATP13A3 WT (A3 WT-OE) or a catalytic dead mutant D498N (A3 D498N-OE), and confirmed that both the ATP13A3 expression level (Appendix A) and endocytosis rate (Appendix A) of A3 WT-OE and A3 D498N-OE HMEC-1 cells were comparable. Hence, differences in BODIPY-polyamine uptake between A3 WT-OE and A3 D498N-OE cells can be attributed to a functional involvement of the transport ATPase.

Next, we analyzed the time- and dose-dependency of BODIPY-SPD and BODIPY-PUT uptake in the ATP13A3 cell models (Figure 3 and Appendix A). First, we evaluated the ATP13A3-dependent BODIPY-PUT and BODIPY-SPD uptake (5 µM) in HMEC-1 over time (Appendix A and Figure 3A,C). At 4 h, the cellular uptake of both probes reached saturation. Interestingly, at 5 µM BODIPY-polyamine, A3 WT-OE cells exhibited a more pronounced uptake than A2 WT-OE cells, already at early time points. Therefore, a shorter incubation time of 30 min was required to remain in the linear uptake phase in A3 WT-OE cells (Figure 3A,C) compared with the 2 h incubation in A2 WT-OE cells (Figure 2A). This may reflect the differences in (i) endosomal localization between ATP13A3 (early/recycling endosomes) and ATP13A2 (late endo/lysosomes) [11,12,13], (ii) cell type properties (endothelial versus neuroblastoma cells) and/or (iii) relative expression levels of ATP13A3 versus ATP13A2.

Second, we performed a dose–response experiment with increasing concentrations of BODIPY-PUT or -SPD (0–100 µM) after 30 min incubation (Figure 3B,D). At 1 µM, ATP13A3-mediated BODIPY-SPD uptake was more pronounced than BODIPY-PUT, indicating that ATP13A3 may present at a higher apparent affinity and/or V_max_ for BODIPY-SPD. A higher maximal fold increase was also observed with BODIPY-SPD (33-fold at 0.1 µM, Appendix A) than for BODIPY-PUT (15-fold at 5 µM, Appendix A). The higher fold uptake with 5 µM ^14^C-SPD (41-fold) (Figure 3E) than with 5 µM ^14^C-PUT (26-fold) (Figure 3F) indicates that ATP13A3 presents a higher transport activity with SPD than with PUT. To investigate the impact of the fluorescent headgroup, we compared the uptake of BODIPY-labeled with ^14^C-radiolabeled SPD and PUT in A3 WT-OE cells at two different concentrations (1 µM and 5 µM) (Figure 3D,E). At 1 µM, a comparable fold change in uptake can be observed between BODIPY-PUT (9.4-fold) versus ^14^C-PUT (11-fold), whereas the uptake of BODIPY-SPD was significantly higher (21-fold) than ^14^C-SPD (11.6-fold). At 5 µM, the uptake of the BODIPY probes was lower compared with ^14^C-labeled polyamines, especially for BODIPY-SPD. Our results therefore suggest that ATP13A3 presents a higher apparent affinity and/or reduced maximal uptake activity toward BODIPY- as compared with ^14^C-labeled probes, similar to what we have observed for ATP13A2.

### 3.3. Chemical Synthesis of Novel Green Fluorescent Polyamine Conjugates

Our results so far demonstrate that both radiolabeled and BODIPY-polyamines are taken up in cells via ATP13A2 and ATP13A3. Next, we explored whether the coupling strategy or the headgroup fluorophore of the fluorescent conjugates may influence the transport by ATP13A2 and/or ATP13A3 (Figure 4). The synthesis of the click-chemistry derived BODIPY-conjugated polyamines (compounds **1a–c** in Figure 4) was performed as previously described [11,21], with small modifications indicated in Appendix A. Here, we report the synthesis of various additional polyamines conjugated to green fluorophore headgroups. First, BODIPY-FL-T-polyamines were synthesized via a similar click-chemistry reaction as for the original BODIPY probes [21], but with the difference that azido-polyamines (-PUT, -SPD or -SPM) were clicked onto the alkyne-BODIPY-FL fluorophore. This resulted in the formation of a triazole ring (T) between the polyamine and the BODIPY-FL groups (referred to as FL-T probe, compounds **2a–c**, Figure 4). Second, a succinimidyl ester–amine coupling strategy yielded the synthesis of BODIPY-FL-A conjugated polyamines in which an amide bond (A) was created between the polyamine and BODIPY-FL groups (referred to as FL-A probe, compounds **3a–c**, Figure 4). Lastly, an isothiocyanate–amine coupling strategy was used to generate FITC-conjugated polyamines with a thiourea bond between the polyamine and the fluorophore (referred to as FITC probes, compounds **4a–c**, Figure 4). The detailed chemical synthesis of each polyamine-conjugate is outlined in Appendix A and the complete chemical structures are drawn in Appendix A. As expected, the fluorescent properties of the different fluorophore-polyamine conjugates (i.e., the wavelengths of maximum absorption and emission λ_Abs_ (max) and λ_Em_ (max), as well as the Stokes shift) are very similar (Appendix A). Note that the linkage to the different fluorophores does not carry a charge and at physiological pH all PUT, SPD and SPM probes are almost fully protonated, sharing close to two, three or four positive charges, respectively [28].

To evaluate whether the new probes are substrates for ATP13A2, we performed ATP/NADH enzyme-coupled ATPase assays on purified ATP13A2 with FL-A-SPM, FL-T-SPM and FITC-SPM (Appendix A) using SPM and BODIPY-SPM as references. ATP13A2′s activity is not only stimulated by SPM and BODIPY-SPM, but also by FL-A-, FL-T- and FITC-SPM, showing that the different fluorophores or coupling strategies do not prevent the recognition of the polyamine core as a substrate. Interestingly, only BODIPY-SPM, and not the other fluorophore-SPM conjugates, displays a higher apparent affinity for ATP13A2, since the new probes stimulate ATP13A2 ATPase activity to the same extent as the unlabeled SPM up to a concentration of 10 µM (Appendix A).

### 3.4. Fluorescent Polyamine Probes Present Different Structure–Function Relationships toward ATP13A2 and ATP13A3

In the next step, we compared the uptake of the new green fluorescent polyamines in both ATP13A2 and ATP13A3 cell models using the original BODIPY-polyamines as a reference (Figure 5A,E). ATP13A3 and ATP13A2 cell models were exposed to a fixed concentration of 5 µM fluorescent polyamine for 30 min or 2 h, respectively (Figure 5). In general, we observed a larger fold-change of uptake with the new FL-A and FL-T probes than with the reference BODIPY polyamines. The large window between uptake in cells expressing WT versus the catalytic dead mutant shows that all probes are mainly transported via the catalytic activity of ATP13A2 or ATP13A3 (Figure 5), in line with the ATPase data for ATP13A2 (Figure 4). Strikingly, the uptake windows obtained with the new BODIPY-FL-probes (FL-A and FL-T) are much larger than with the reference BODIPY polyamines. The structural difference lies not only in the substitution pattern on the fluorophore core, but also on the position of the connection toward the polyamine. Both of these factors may affect the cellular uptake via ATP13A2 and/or ATP13A3. Despite their lower apparent affinity for the transporter compared with the original BODIPY analogs, the signal to noise ratio for the uptake was highest with the FL-A probes, followed by the FL-T probes.

In line with the biochemically confirmed substrate specificity, A2 WT-OE cells presented a significantly higher uptake of all SPD and SPM conjugates. A more modest, but significant uptake of the BODIPY-PUT analogs was observed, whereas uptake of FITC-PUT was not significantly different from A2 D508N-OE cells (Figure 5D,H). Conversely, A3 WT-OE cells exhibited a higher uptake of BODIPY-labeled PUT and SPD probes than of the fluorescent SPM conjugates (significantly different only for FITC-SPM and FL-A-SPM). The fluorophore FITC seemed to prevent PUT recognition in ATP13A3 cells (Figure 5H). This is remarkable, because ATP13A3 accepts all other fluorophore-PUT conjugates reported here (Figure 3E,F and Figure 5E–G), as well as ^14^C-PUT (Figure 3E,F). Notably, FL-A-SPD offers the best uptake window for both ATP13A2 (14-fold) and ATP13A3 (105-fold), and therefore emerges as the best probe to follow either ATP13A2 or ATP13A3 transport activity in cells. Conversely, FL-T-PUT exhibits the highest specificity for ATP13A3 (35-fold more uptake in ATP13A3 versus ATP13A2 cells), whereas FITC-SPM presents the highest specificity for ATP13A2 (1.8-fold more uptake in ATP13A2 versus ATP13A3 cells). Together, our data show that ATP13A2 and ATP13A3 exhibit overlap in substrate specificity, whereas a probe-specific structure–activity relationship (SAR) toward ATP13A2 and ATP13A3 has been observed (Figure 5F,G).

## 4. Discussion

Fluorescently labeled polyamine conjugates represent powerful tools to assess polyamine transport at the cellular level and dissect the mPTS that remains poorly characterized. In this study, we compared the properties of different green fluorescently labeled polyamines to analyze the transport activity in cells of the two housekeeping P_5B_-type ATPases, ATP13A2 and ATP13A3, which emerge as major determinants of cellular polyamine uptake [11,13].

### 4.1. Comparison of ATP13A2 and ATP13A3 Substrate Specificity

We provided the first biochemical evidence that fluorescently labeled polyamines stimulate the catalytic turnover of ATP13A2, indicating that they are genuine substrates of ATP13A2. We previously documented that fluorescent polyamine conjugates enter cells via endocytosis [13] and are subsequently transported via the activity of ATP13A2 from late endo/lysosomes to the cytosol and further into other compartments, including mitochondria [13,23]. We also showed that ATP13A2 activity responds to a range of fluorescent polyamine analogs, indicating flexibility of the substrate binding pocket. We observed the best correlation between the cellular uptake levels of BODIPY-FL-T labeled polyamines (BODIPY-FL-T SPM > SPD > PUT) and the reported apparent affinity of ATP13A2 for unlabeled polyamines (SPM > SPD > PUT; [13]). This correlation is weaker for other polyamine conjugates, indicating that amongst the here tested fluorophore polyamine analogs BODIPY-FL-T probes may accommodate best to the substrate binding pocket.

The radio- and fluorescently labeled polyamine uptake data for ATP13A3 point to PUT and SPD as preferred substrates of the protein, but not all polyamine probes provide the same results. All fluorescent SPD analogs and the radiolabeled SPD probe are taken up via ATP13A3, indicating that SPD truly represents a transported substrate. ATP13A3 also promotes the uptake of radiolabeled PUT, which is in line with the strong uptake of PUT coupled to BODIPY, BODIPY-FL-A or BODIPY-FL-T. These data are in good agreement with our previous study in CHO-MG cells, where ATP13A3 complements PUT uptake deficiency [11]. However, no ATP13A3-dependent uptake was observed with FITC-PUT, suggesting that the FITC-label disrupts PUT recognition by ATP13A3. Conversely, only FITC-SPM, and not other SPM analogs, are taken up in cells via ATP13A3, making it less likely that SPM represents a transported substrate of ATP13A3. This contrasts with a recent study that highlighted both SPD and SPM, but not PUT, as likely ATP13A3 substrates based on radiolabeled uptake experiments in pancreatic cancer cells [15]. On the other hand, in CHO-MG cell models, we previously observed that unlabeled PUT, SPD and SPM competed with BODIPY–PUT uptake via ATP13A3, indicating that PUT, SPD and SPM may all be transported by ATP13A3 [11]. The apparent discrepancy between these studies may point to cell-type specific properties of ATP13A3 or may reflect differences in the experimental conditions, such as exposure time, fluorescent polyamine concentrations or cell culture conditions. Based on the uptake data, it remains challenging to pinpoint the precise endogenous substrates of ATP13A3, and biochemical confirmation on purified ATP13A3 will be required to establish the relative affinities of ATP13A3 toward the endogenous polyamine species.

The overlapping substrate specificities suggest that ATP13A2 and ATP13A3 fulfill—at least in part—redundant functions in cellular polyamine uptake. This has been confirmed in CHO-MG cells with loss of ATP13A3 functionality that can be fully rescued not only with ATP13A3, but also with ATP13A2 complementation [11]. It remains unclear whether the redundancy of the isoforms would play a compensatory or maladaptive role in diseases associated with ATP13A2 or ATP13A3 mutations.

Overall, our series of green fluorescent probes are useful tools to assess ATP13A2 and ATP13A3 activity in cells, but not to deduce the endogenous substrate specificities. While both isoforms promote the uptake of all SPD labeled analogs, FL-A-SPD offers the best uptake window. FL-A-SPD therefore emerges as an excellent probe to follow the combined cellular activity of ATP13A2 and ATP13A3 with high sensitivity and uptake potential.

### 4.2. Toward the Design of ATP13A2 and ATP13A3 Specific Polyamine Probes

ATP13A2 and ATP13A3 both transport SPD, pointing to an overlap in substrate specificity. However, the P_5B_ isoforms present differences in their activity toward PUT or SPM analogs, since the fluorophore and coupling strategies differently affect the uptake of the probes in ATP13A2 versus ATP13A3 cell models. So far, FL-T-PUT emerges as the best polyamine analog to probe ATP13A3 activity selectively, whereas FITC-SPM may be considered to more selectively follow ATP13A2 activity. Previously, we described that ATP13A3, but not ATP13A2, mediates cellular toxicity to methylglyoxal bis-(guanylhydrazone) (MGBG), a toxic SPD analog and an inhibitor of polyamine biosynthesis [11]. The overlapping, as well as isoform-specific substrate recognition of ATP13A2 versus ATP13A3, is compatible with a strong, but not complete, conservation of the substrate binding pocket between P_5B_ isoforms. Indeed, most critical residues for polyamine binding and recognition are conserved between ATP13A2 and ATP13A3 [19], allowing that the polyamine part of the fluorescent probe may enter the channel-like polyamine binding pocket regardless of the linker or fluorophore. Polyamines bind in their linear form and the polyamine affinity depends on the number of amine groups and the spacing between them [19]. The polyamine part of the fluorescent probes may subsequently translocate within the membrane domain of the transporter following an unresolved path. Since the polyamine binding site may be too narrow to accommodate a bulky fluorescent headgroup, we hypothesize that the fluorescent tag may instead move along/in between transmembrane helices or—given its hydrophobic nature—may stick out within the membrane bilayer. The translocation mechanism of fluorescent polyamines may resemble the ‘credit card swipe’ mechanism described in lipid flippases of the P_4_-type ATPase family. An entire phospholipid substrate does not fit within the central transmembrane domain of the flippase; hence, only the polar headgroup region of the phospholipid interacts with the protein, while the fatty acyl tails slide through the hydrophobic region of the lipid bilayer membrane surrounding the protein [29]. A similar translocation mechanism for the fluorescently labeled polyamines may explain why ATP13A2 and ATP13A3 translocate fluorescent polyamine analogs with different fluorophores and linkers. The isoform-specific amino-acid differences in the proximity of the polyamine binding site may contribute to the SAR by influencing the passage of the bulky fluorescent label. The higher apparent affinity of ATP13A2 for BODIPY-SPM as compared with SPM or other fluorescent SPM probes may therefore be possibly explained by specific interactions between BODIPY and surrounding residues in the transmembrane domain. Besides BODIPY- and FITC-labeled polyamines, nitrobenzoxadiazolyl (NBD)- [30], anthracene- [31] and indotricarbocyanine-labeled polyamines [32] have also previously been developed to monitor the activity of the mPTS. Additionally, the NBD-labeled spermine is most likely taken up via ATP13A3 [15], but it remains to be tested whether all these probes are indeed transported via ATP13A2 and/or ATP13A3.

### 4.3. Distinct Polyamine Uptake Kinetics in ATP13A2 and ATP13A3 Cells

We found that ATP13A3 mediates a more prominent and faster uptake of radio- and fluorescently labeled polyamines in HMEC-1 cells than does ATP13A2 in SH-SY5Y cells. Indeed, ATP13A3 contributes more to cellular PUT and SPD uptake than does ATP13A2. ATP13A3 may also contribute more to cellular SPM uptake, since the uptake of fluorescent SPM analogs in ATP13A2 and ATP13A3 cells is comparable despite the shorter treatment time in ATP13A3 models. The expression of ATP13A3 in earlier compartments of the endosomal system, as well as specific biochemical properties (K_m_, V_max_), may explain the more dominant and faster polyamine uptake via ATP13A3 over ATP13A2, although further investigations will be required to confirm whether the differential impact of ATP13A2 and ATP13A3 on polyamine uptake remains valid in other cell types.

Our results show that with a proper choice of the polyamine probe, concentration and short treatment time, it is possible to exclusively measure ATP13A3 dependent uptake in cells expressing both ATP13A2 and ATP13A3. However, it remains much more challenging to dissect the ATP13A2 activity in the background of ATP13A3, since ATP13A2-specific probes have not yet been identified. In addition, endocytosed probes first meet ATP13A3 in the earlier endosomal compartments from which they can be exported by ATP13A3 before reaching ATP13A2 in the late endo/lysosomal compartments.

Importantly, the distinct substrate specificities and uptake kinetics of ATP13A2, as well as the contribution of the fluorophore and linker to substrate recognition and affinity, indicate that designing ATP13A2-specific probes may be feasible. With isoform-specific probes we would be able to follow transporter-specific uptake in various cell types and physiological conditions. In disease models we may pick up isoform-specific transport deficiencies and map possible compensatory changes in the activity of other P_5B_ ATPase isoforms. Moreover, the broad range of fluorescent polyamines that enter cells via ATP13A2 and/or ATP13A3 suggest that polyamines may be considered as vehicles to take up drugs or other molecular cargo [33]. For instance, polyamine–anthracene conjugates have previously been designed to target cancer cells with an increased polyamine uptake system [34] or *Plasmodium falciparum* parasites [35]. Structural studies of P_5B_-ATPases with bound polyamine conjugates as well as functional analysis of other polyamine conjugates will lead to a better understanding of the SAR, which will facilitate the design of isoform-specific strategies.

### 4.4. Green Fluorescent Probes with Improved Synthesis and Properties

The novel green fluorescent polyamine probes described here follow two different synthetic strategies. Our original synthesis encompassed a click-chemistry-based approach between an azide-functionalized polyamine and an alkyne-labeled BODIPY fluorophore. Here, we also explored amide and isothiocyanate coupling to a properly protected polyamine analog with one single free amino group. Conveniently, such protected polyamines are intermediates in our original probe synthesis, and therefore shorten the route by one step. We found that the amide coupling strategy gave rise to the most promising fluorescent polyamine series (BODIPY-FL-A series; **3a–c**). As many more fluorophore oxysuccinimide esters are available from commercial sources, we expect that other analogs can be readily synthesized, opening the way to further improvement of the fluorescent polyamine properties.

Here, we found that the new probes, despite having a lower affinity than our previously reported fluorescent polyamine probes, display a higher cellular uptake at 5 µM. This indicates that the new probes (except for FITC-PUT, **4c**) are superior substrates for ATP13A2 and ATP13A3, and underlines the importance of the nature of the fluorophore and the coupling strategy between the fluorophore and the polyamine.

In conclusion, the here reported fluorescent polyamine probes are attractive for the study of polyamine uptake by ATP13A2 and ATP13A3, and the BODIPY-FL-A series emerged as the most efficient of the tools described here. 

## Figures and Tables

**Figure 1 biomolecules-13-00337-f001:**
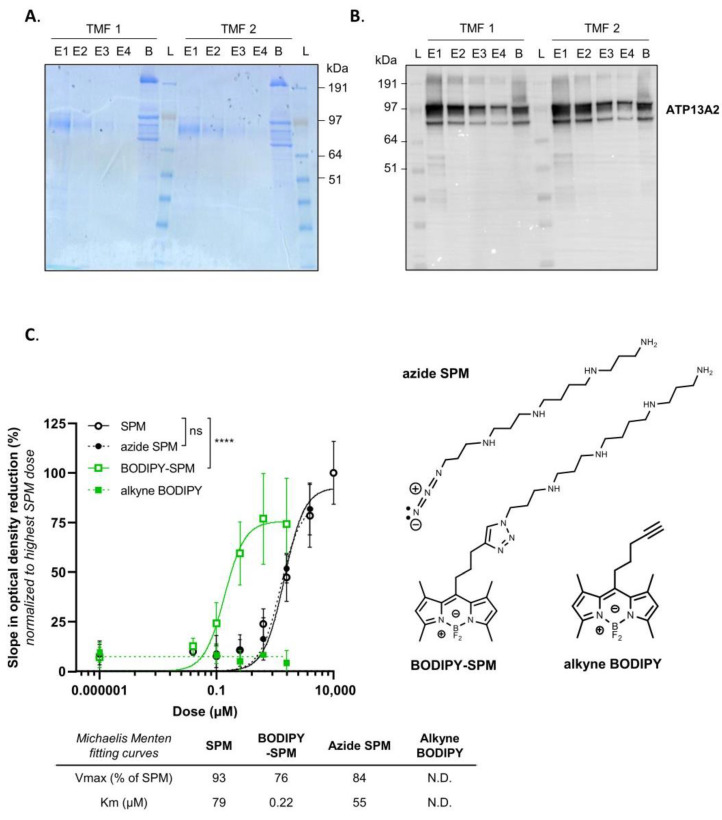
Purification and ATPase activity of hATP13A2. (**A**). Coomassie staining showing the purified human ATP13A2 protein present in the different elution fractions (E1 to E4) from total membrane fraction (TMF) and onto the beads (B). (**B**). Immunoblot analysis of ATP13A2 purified protein from the fractions depicted in (**A**,**C**). ATPase activity of purified ATP13A2 (1.25 µg) measured with increasing concentrations of unlabeled spermine (SPM), clickable spermine (azide SPM), free BODIPY (alkyne BODIPY) and BODIPY-SPM, which structures are presented on the right (N = 2 to 7, technical duplicates, two-way ANOVA with Tukey’s multiple comparisons test; ns: not significant; ****: *p* < 0.001). The Y axis depicts the slope in optical density at 340 nm reflecting NADH consumption and further ATP consumption, normalized to the highest dose of SPM. V_max_ and K_m_ shown in the table were determined using the Michaelis–Menten fitting in GraphPad Prism software. N.D.— not determined.

**Figure 2 biomolecules-13-00337-f002:**
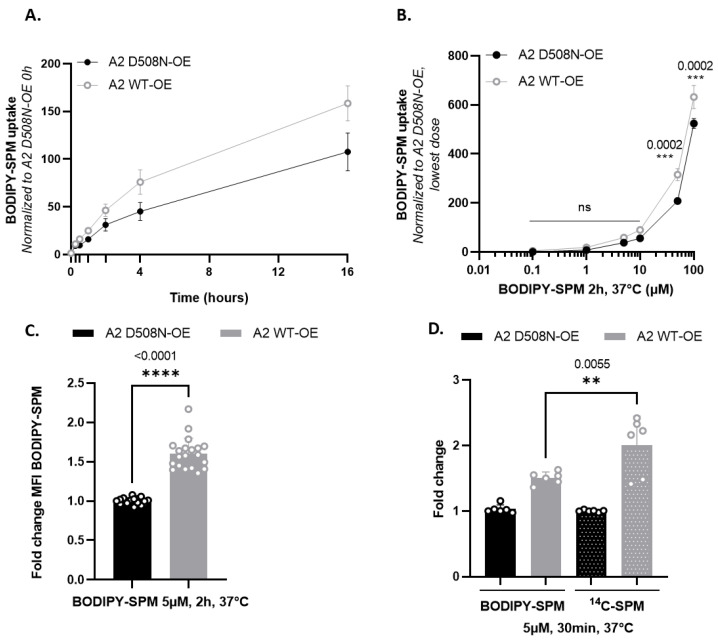
Time- and dose-dependency of BODIPY-SPM uptake in ATP13A2-expressing SH-SY5Y cell models. (**A**,**B**). SH-SY5Y cells were incubated with 5 µM BODIPY-SPM for the indicated times ((**A**), time dependency) or with increasing concentrations of BODIPY-SPM for 2 h ((**B**), dose dependency). The mean fluorescence intensity (MFI) of 10,000 events was recorded using a BD Canto II HTS flow cytometer. Graphs represent the MFI recorded in ATP13A2 WT OE (A2 WT-OE) and ATP13A2 D508N-OE (A2 D508N-OE) cells, normalized to the 0 h incubation time in A2 D508N-OE cells (in **A**) or to the lowest concentration of BODIPY-SPM (i.e., 100 µM) in A2 D508N-OE cells (in **B**; ns: not significant, **: *p* < 0.01; ***: *p* < 0.001; ****: *p* < 0.0001). (**C**)**.** Graph depicts the fold change in BODIPY-SPM uptake (5 µM; 2 h) between A2 WT-OE and A2 D508N-OE cells (N = 10, technical duplicates, unpaired t-test). (**D**)**.** Comparison of the uptake of 5µM BODIPY-conjugated (BDP-SPM) versus 5 µM radiolabeled spermine (^14^C-SPM) in SH-SY5Y A2 WT-OE and D508N-OE cells for 30 min (N = 3, technical duplicates, one-way ANOVA with Tukey’s multiple comparisons test).

**Figure 3 biomolecules-13-00337-f003:**
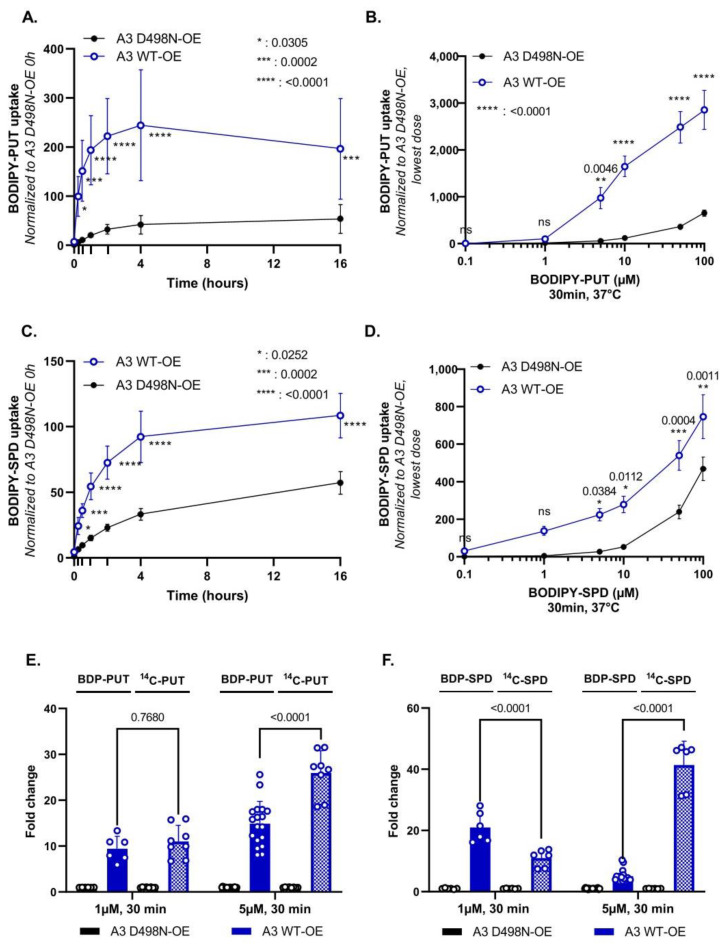
Time- and dose-dependency of BODIPY-PUT and BODIPY-SPD uptake in ATP13A3-expressing HMEC-1 cell models. (**A**,**C**). Time dependent BODIPY-polyamine uptake in HMEC-1 cells incubated with 5 µM BODIPY-PUT (**A**) or BODIPY-SPD (**C**). The mean fluorescence intensity (MFI) of 10,000 events was recorded using a BD Canto II HTS flow cytometer. Graphs represent the fold change between the MFI recorded in A3 WT-OE versus A3 D498N-OE cells, normalized to the 0 h incubation time in A3 D498N-OE cells. (N = 3 to 9, with technical duplicates, two-way ANOVA with Sidak’s multiple comparisons test). (**B**,**D**). Dose–response of BODIPY-PUT (**B**) or BODIPY-SPD (**D**) uptake in HMEC-1 cells at 30 min. The MFI of 10,000 events was recorded using a BD Canto II HTS flow cytometer. Graphs represent the fold change difference between the MFI recorded in A3 WT-OE and A3 D498N-OE cells, normalized to the highest concentration of BODIPY-polyamine (i.e., 100 µM) in A3 D498N-OE cells (*N* = 3 to 9, with technical duplicates, two-way ANOVA with Sidak’s multiple comparison test; ns: not significant, *: *p* < 0.1; **: *p* < 0.01; ***: *p* < 0.001; ****: *p* < 0.0001). (**E**,**F**). Comparison of the uptake of 1 and 5 µM BODIPY-conjugated (BDP-PUT and BDP-SPD) versus radiolabeled putrescine and spermidine (^14^C-PUT and ^14^C-SPD) in HMEC-1 A3 WT-OE and D498N-OE cells for 30 min (N = 3 to 9, technical duplicates, two-way ANOVA with Tukey’s multiple comparisons test).

**Figure 4 biomolecules-13-00337-f004:**
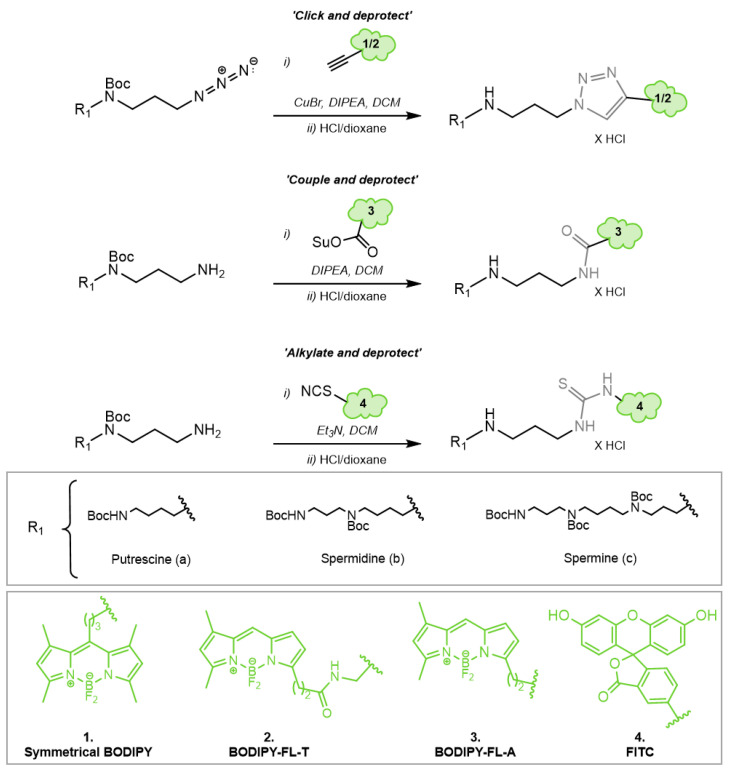
Simplified scheme representing the chemical synthesis of the different green fluorescent polyamine conjugates used in the study. A detailed description of the synthesis is given in Appendix A.

**Figure 5 biomolecules-13-00337-f005:**
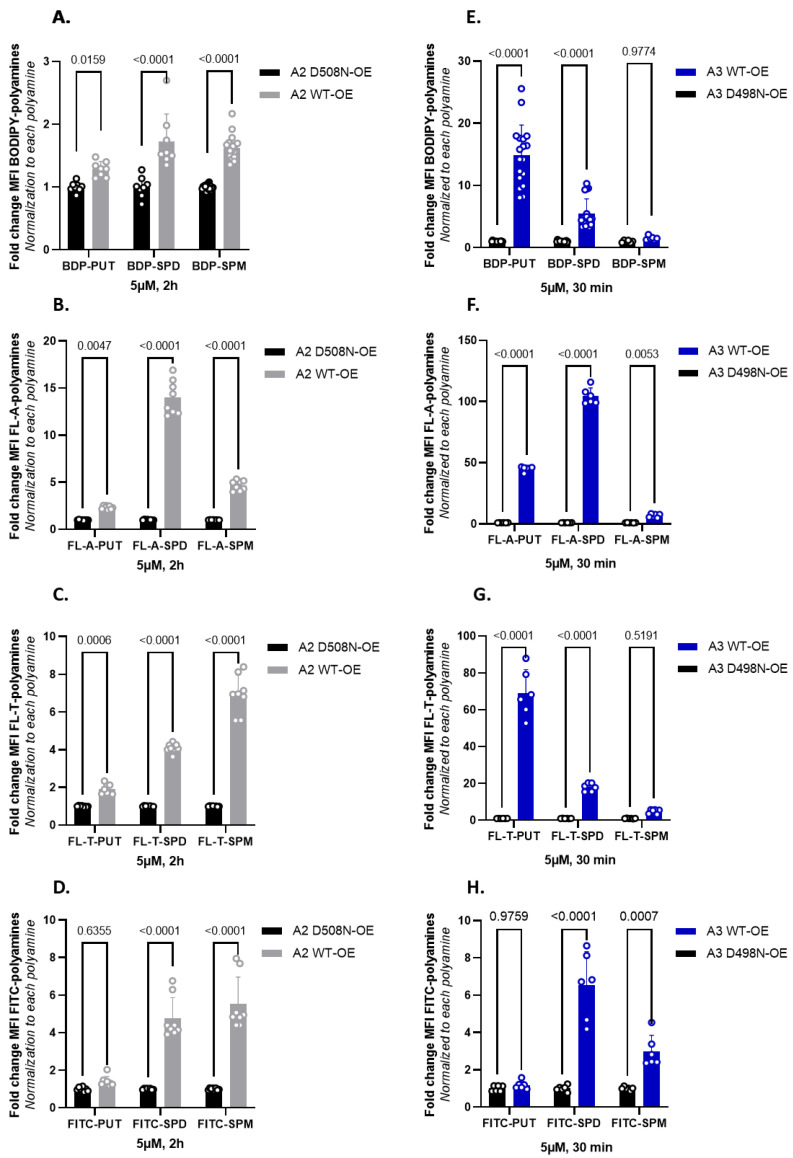
Comparison of the cellular uptake of BODIPY-conjugated versus BODIPY-FL-A, BODIPY-FL-T and FITC-conjugated polyamines in ATP13A2 and ATP13A3 cell models. (**A**–**D**). SH-SY5Y, ATP13A2, WT-OE and D508N-OE cells were incubated for 2 h with 5 µM of each fluorescently labeled polyamine. (**E**–**H**). HMEC-1, ATP13A3, WT-OE and D498N-OE cells were incubated for 30 min with 5 µM of each fluorescently labeled polyamine. The mean fluorescence intensity (MFI) was recorded using a BD Canto II flow cytometer based on the settings used for BODIPY-polyamines and normalized as indicated on the Y axis. Experiments were performed four independent times (N = 4) in ATP13A2 cell model and three independent times (N = 3) in ATP13A3 cell model, with technical duplicates. Statistical analysis was conducted using GraphPad Prism and a two-way ANOVA test with Sidak’s multiple comparison test.

## Data Availability

All datasets generated or analyzed in this study can be found through the Zenodo depository, reserved doi: 10.5281/zenodo.7434965. All experimental protocols can be found on protocols.io.

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
