# Peer review of "Novel Green Fluorescent Polyamines to Analyze ATP13A2 and ATP13A3 Activity in the Mammalian Polyamine Transport System"

_biomolecules, 2023, doi:10.3390/biom13020337_

Round 1

Reviewer 1 Report

minor typos and comments:

line 43: flies

line 46: system

line 67: insert a comment about how the mode of attachment of the BODIPy probe affects the charge of the molecule at physiological pH. Amide linker is neutral but is the Triazole linker protonated at pH 7.4? 

line 91: does the DMSO vehicle impact the assay? Control needs to be mentioned or shown. 

line 125: the paper needs a control run with aminoguanidine and at least one probe ( Bodipdy FL Spd for example) to show that amine oxidases do not impact the assay. Line 205 suggests a 16h incubation period which may introduce artifacts from amine oxidases present in FBS? Thus one needs a control to show this is it an issue here. 
line 188: insert prose stating that the cells were washed here, no?

line 211 could one look by plate reader at the washed cells for probe uptake directly rather than by flow? If so that would simplify the assay steps.

how were the Vmax and Ki determined.  Is there a ref for that?

line 271: what was the relative ATP13A3 exp? Was this considered?

Fig S2 legend . Pitstop2

line 433: insert a mention about the number of charges expected for each probe at physiologic pH  

lines 572 and 602: anthracene 

line 618: consider mentioning the new commercially available Polyamine Red probe  

line 626: ATP13A3

Refs: 

several refs have et al listed. Check whether the journal needs the full reference listed including full page ranges for each ref.

lines 698 and 736 and 747 should be not have the ‘t’. after Phanstiel, O.

line 706: KPK? Is that correct?

line 725: remove brackets around ATP13A3

Overall the paper is well presented and makes a significant contribution in this area. Prior work is appropriately cited as well as the authors insights into how these may target one or both of the ATPases.

While some of the methods employed here like ATPase activity provide indirect measures of the ATPase driven polyamine uptake process there are enough data here to support the authors’ conclusions and I suggest publishing this work after the authors address my minor comments above. The aminoguanidine expt control is a must to place this work in context with other publications in this area. I congratulate the authors for their efforts to better explain this challenging area.

Author Response

Please see in attachment. 

Reviewer 2 Report

The manuscript titled "Novel green fluorescent polyamines to analyze ATP13A2 and ATP13A3 activity in the mammalian polyamine transport system" by Houdou and colleagues describes the generation and characterization of polyamine derivatives for use in studying the enigmatic mammalian polyamine transport system. Using cell- and recombinant protein-based techniques, the authors evaluate substrate preferences for the recently described ATP13A2 and ATP13A3 polyamine transport-associated proteins, comparing their novel fluorescent analogues with previously developed BODIPY-labeled polyamines as well as radiolabeled polyamines traditionally utilized in transport studies. The manuscript is very well written and organized, and the results advance the current knowledge of polyamine transport as well as provide valuable experimental tools for future studies of mammalian polyamine homeostasis. I have only a couple of very minor suggestions:

Line 141: "scrapping" should be "scraping"

Line 191: reference to SI Figs S5-8 precedes reference to SI Figs S1-4 (line 272)

Line 273: define "NTS" cells

Section 4.3, 1st paragraph: I feel that more caution should be taken in making these comparisons, as there are often very large differences in polyamine uptake kinetics among different cell types, as well as the potential for variable levels of overexpression. These data could be said to suggest that ATP13A3-mediated uptake is faster/more dominant than ATP13A2, but additional confirmation is required in a system with less variables.

Round 2

Reviewer 1 Report

Line 142: Others have used exogenous aminoguanidine for this purpose. (add new REF:

Autophagy induction by exogenous polyamines is an artifact of bovine serum amine oxidase activity in culture serum, Cassandra E. Holbert et al, J Biol Chem, 2020. Volume 295, ISSUE 27, P9061-9068.

Overall probe charge is critical to uptake and recognition. The probes used differ in the number of charges presented at physiological pH. For example, spermidine is 93% protonated at physiological pH, so these probes are more likely 'mostly protonated' but not completely protonated species at pH 7.4.

Ref: Measurement of Polyamine pKa Values by Ian S. Blagbrough, Abdelkader A. Metwally, and Andrew J. Geall published in Anthony E. Pegg and Robert A. Casero, Jr. (eds.), Polyamines: Methods and Protocols, Methods in Molecular Biology, 2011,720, pp. 493-503.
DOI 10.1007/978-1-61779-034-8_32

line 434: While the amide and thiourea linkages do not have formal charges, the 1,2,3-triazole link has a pKa over 9.3 and should be protonated as well at physiological pH. Thus the BoDIPY-FL-A and FITC derivatives are more chemically similar to the native polyamines, as they represent N1-substituted native polyamine structures, and have the same number of charges to present to the ATPase polyamine recognition sites. Please insert a sentence pointing out this distinction to the reader (i.e., triazoles are expected to have an additional charge due to the linkage) and as such the triazoles may behave differently based on charge differences.

Line 504: insert a brief comment that the presence of the triazole may impact this outcome by providing a third charge which causes the molecule to look more like spermidine rather than putrescine.

Other than these minor points the authors have done a good job revising the manuscript.
